# Persistent dynamic magnetic state in artificial honeycomb spin ice

J. Guo[1,4], P. Ghosh[1,4], D. Hill[1,4], Y. Chen[2], L. Stingaciu [3], P. Zolnierczuk [3], C. A. Ullrich [1] ✉ & D. K. Singh [1] ✉

Topological magnetic charges, arising due to the non-vanishing magnetic flux on spin ice vertices, serve as the origin of magnetic monopoles that traverse the underlying lattice effortlessly. Unlike spin ice materials of atomic origin, the dynamic state in artificial honeycomb spin ice is conventionally described in terms of finite size domain wall kinetics that require magnetic field or current application. Contrary to this common understanding, here we show that a thermally tunable artificial permalloy honeycomb lattice exhibits a perpetual dynamic state due to self-propelled magnetic charge defect relaxation in the absence of any external tuning agent. Quantitative investigation of magnetic charge defect dynamics using neutron spin echo spectroscopy reveals sub-ns relaxation times that are comparable to the relaxation of monopoles in bulk spin ices. Most importantly, the kinetic process remains unabated at low temperature where thermal fluctuation is negligible. This suggests that dynamic phenomena in honeycomb spin ice are mediated by quasi-particle type entities, also confirmed by dynamic Monte-Carlo simulations that replicate the kinetic behavior. Our research unveils a macroscopic magnetic particle that shares many known traits of quantum particles, namely magnetic monopole and magnon.

Magnetic structure and dynamics in a magnetic material are governed by the underlying energetics due to the interaction of magnetic moments with external and internal magnetic fields[1,2]. Unlike its classical counterpart, a quantum magnet manifests a persistent dynamic tendency associated with spin or magnetic moment relaxation[2,3]. A magnetic moment can be thought of as a pair of + and − quasi-particles, called magnetic charges, coupled via magnetic Coulomb interactions[4]. This concept plays a dominant role in the temperature-dependent evolution of magnetic and physical properties in spin ice compounds[5–7]. A direct demonstration of this effect is most discernible in two-dimensional artificial kagome ice on a honeycomb motif, where the possible local moment configurations (2-in and 1-out, 1-in and 2-out, and all-in or all-out) impart low ($Q$) and high ($3Q$) multiplicity magnetic charges on respective vertices[8–11]. Unequivocal evidence of magnetic charges has been obtained using magnetic force microscopy in

honeycomb lattices made of permalloy ($Ni_{0.81}Fe_{0.19}$) and cobalt magnets[12–15]. However, they are not known to be dynamic in the absence of external tuning agents (magnetic, thermal, or current)[8,9,15,16], which is in strong contrast to the well-established dynamic properties of magnetic charges in analogous spin ice compounds that exhibit persistent or continuous relaxation at low enough temperature[6,17,18]. Magnetic charges relax by emitting or absorbing net charge defects, $q_m$, of magnitude $2Q$, also termed effective monopoles[2,4,8,12,19].

A key question is whether magnetic charge in artificial spin ice can be described by the same quantum mechanical treatment as in spin ice, where the Hamiltonian representing magnetic charge interaction is defined by Pauli matrices[3,4]. A direct comparison of magnetic charge relaxation times in spin ice and artificial magnetic honeycomb lattices can shed light on this fundamental problem. From the Coulombic physics perspective, a magnetic charge at the center of an artificial

[1]Department of Physics and Astronomy, University of Missouri, Columbia, MO, USA. [2]Suzhou Institute of Nano-Tech and Nano-Bionics, Chinese Academy of Sciences, Suzhou, China. [3]Oak Ridge National Laboratory, Oak Ridge, TN 37831, USA. [4]These authors contributed equally: J. Guo, P. Ghosh, D. Hill. ✉e-mail: ullrichc@missouri.edu; singhdk@missouri.edu

kagome lattice can be mapped onto the net magnetic flux along the bond direction on the parent honeycomb motif[20,21]. The correspondence entails the quantum treatment of magnetic charge interactions, which has been used previously to deduce thermodynamic and magnetic properties of the statistical ensemble[20,22]. A charge defect's motion between vertices would flip the microscopic moment in the nanoscopic honeycomb element, manifesting a magnonic pattern, thereby altering the moment direction and the net charge on a given vertex, see Fig. 1a–c. The charge defect or magnonic charge, $q_m$, cannot pass through a high integer charge ($3Q$ or $-3Q$), which serves as a roadblock. At the same time, $\pm 3Q$ charges have a high energy cost, which makes them unstable and triggers the dynamic process. A thermally tunable honeycomb lattice, made of ultra-small (-11 nm) nanoscopic permalloy element (see Fig. 1d)[23], can host such dynamic events without the application of any external stimulus. It is known to exhibit a highly disordered ground state, comprised of both $\pm Q$ and $\pm 3Q$ charges, at low temperature[24]. The presence of energetic $\pm 3Q$ charges at low temperatures suggests that the system maintains a high energy state throughout. The ultra-small element size imparts thermally tunable energetics due to the modest inter-elemental magnetic dipolar interaction energy of -45 K.

We use neutron spin echo (NSE) measurements to obtain direct evidence of self-propelled relaxation of magnonic charge $q_m$, as it provides an accurate quantitative estimation of relaxation properties across a broad dynamic range[25,26]. Furthermore, NSE spectroscopy is carried out on two stacks of permalloy honeycomb samples with varying thicknesses of 6 nm and 8.5 nm that render a comprehensive outlook.

## Results

### Neutron spin echo measurements of magnetic charge relaxation

In Fig. 2a, we show the color plot of NSE data obtained on a 6 nm thick honeycomb lattice with net neutron spin polarization along the +Z direction. No meaningful spectral weight is detected in the spin-down neutron polarization (see Supplementary Fig. S4), which confirms the magnetic nature of the signal. To ensure that the observed signal is magnetic in origin, the NSE spectrometer is modified by removing the flipper before the sample (see Methods for detail). The figure exhibits very bright localized scattering pixels that are identified with $q = 0.058\,\text{Å}^{-1}$ and $0.029\,\text{Å}^{-1}$. These $q$ values correspond to the typical relaxation length associated with the distance between the nearest and the next-nearest vertices, $2\pi/l$ and $2\pi/2l$, respectively, with $l = 11$ nm. This immediately suggests the quantized longitudinal nature of magnetic charge defect dynamics in artificial honeycomb lattices. NSE spectroscopy is a quasi-elastic measurement technique where the relaxation of a magnetic specimen is decoded by measuring the

relative change in the scattered neutron's polarization via the change in the phase current at a given Fourier time (related to neutron precession). The localized excitations exhibit pronounced sinusoidal spin echo. Figure 2b, c shows the echo profiles obtained on honeycomb samples at room temperature at 0.02 ns Fourier time, where a strong signal-to-background ratio is detected.

Typically, the echo intensity from a single wavelength $\lambda$ follows a cosine function, given by $I(\phi) = A\cos(\phi\lambda) + B$, where the phase $\phi$ is directly related to the phase current $dJ$. However, if the neutron has a wavelength span of $\lambda_{avg} \pm d\lambda$, then the signals from all different wavelengths add up (see Supplementary Methods C). In this case, the total intensity follows from the functional relation[27]

$$I(\phi) = A\cos(\phi\lambda_{avg})\frac{\sin(\phi d\lambda)}{\phi d\lambda} + B. \qquad (1)$$

As the Fourier time associated with neutron precession increases, the error bar tends to become larger. Figure 2d shows the spin echo at 0.26 ns Fourier time (see Supplementary Fig. S6 for other Fourier times). This trend hints at the fast relaxation of the magnetic charge quasi-particles, which is more prominent at a sub-ns time scale.

The sinusoidal nature of spin echo is quite evident at other temperatures as well. In Fig. 2e, f, we show the plot of characteristic spin echo profiles at low temperatures in both thicknesses at $q = 0.058\,\text{Å}^{-1}$. The strong spin echo at low temperatures suggests the persistence of significant charge dynamics in the nanoscopic system. The origin of charge relaxation at a low temperature of $T = 4$ K is unlikely to be thermal in nature, considering that it costs $|E_Q - E_{-Q}|$ (-76 K) to create a charge defect. Unlike other nanostructured magnets where the domain wall motion, typically ascribed to the dynamic property, is detectable in applied magnetic field or current only, the absence of any external tuning parameter at low temperature suggests the non-trivial nature of magnonic charge $q_m$ relaxation. In total, measurements were performed at six temperatures, $T = 4, 20, 50, 100, 200,$ and 300 K (see Supplementary Fig. S7). The quantitative estimation of relaxation times at different temperatures sheds light on the dynamic state of the artificial honeycomb lattice.

The estimation of the magnetic charge relaxation time is achieved by analyzing the intermediate scattering function $S(q,t)/S(q,0)$ at different spin echo Fourier times. We plot the normalized intensity as a function of neutron Fourier time at $q = 0.058\,\text{Å}^{-1}$ in the 6 nm thick sample in Fig. 3a. Normalization is achieved by dividing the observed oscillation amplitude by the maximum measurable amplitude (see Supplementary Notes B), which is a common practice in magnetic systems with unsettled fluctuations down to the lowest measurement

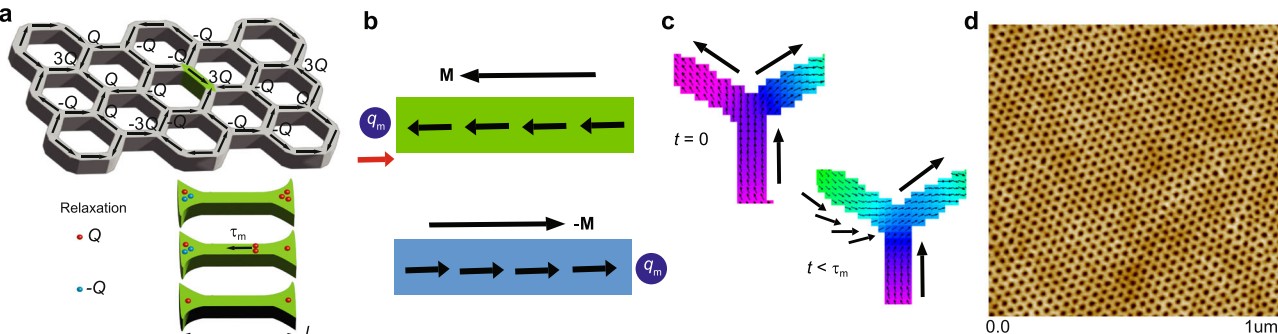

**Fig. 1 | Self-propelled magnetic charge defect relaxation in an artificial honeycomb lattice made of ultra-small permalloy elements. a** Schematic illustration of magnetic charge ($\pm Q$, $\pm 3Q$) relaxation processes between honeycomb vertices. The charge defect ($q_m$) dynamics between nearest neighboring vertices corresponds to the reciprocal wave vector $q$ - 0.058 Å$^{-1}$. Magnetic charge relaxation occurs simultaneously in all three elements attached to a vertex over the relaxation time $\tau_m$. **b** The moment **M** along the honeycomb element changes direction due to the

$q_m$ kinetics. As charge defect traverses between the neighboring vertices, net magnetic charges on the associated vertices change due to the moment flipping. **c** Time-dependent micromagnetic simulations depict magnetization patterns in a honeycomb element at two different instances, $t = 0$ and $t = 363$ ps, of $q_m$ propagation. **d** Atomic force micrograph of artificial honeycomb lattice, with a typical element size of -11 nm (length) × 4 nm (width) × 6 nm (thickness).

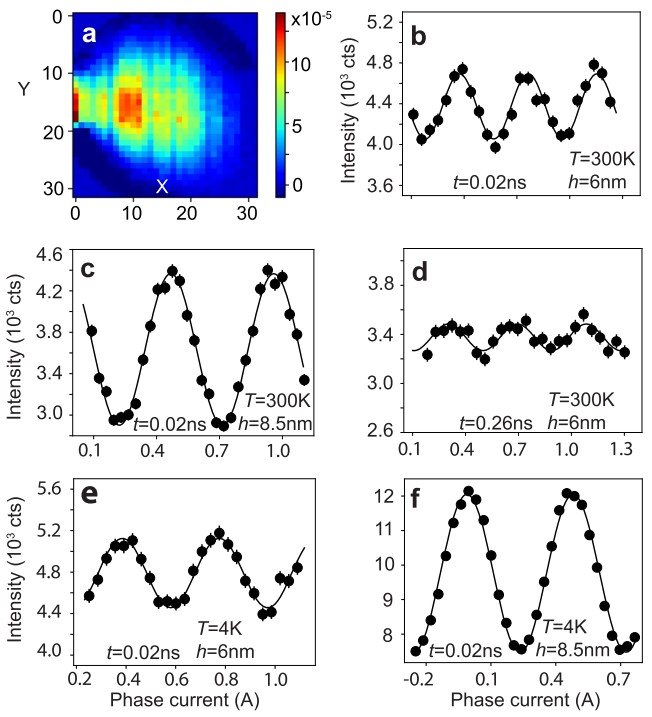

**Fig. 2 | Neutron spin echo spectroscopy revealing dynamic properties in artificial honeycomb lattice without any external tuning parameter. a** Color map of NSE data on a 6 nm thick honeycomb lattice with net neutron spin polarization $(+Z)-(-Z)$. Bright localized scattering pixels are identified with $q = 0.058$ Å$^{-1}$ and $q = 0.029$ Å$^{-1}$ (only partially visible near $x = 0$ due to the geometrical limit of the instrument). NSE data is obtained on a large parallel stack of 125 (117) honeycomb samples of $h = 6$ nm (8.5 nm) thickness. **b, c** Spin echo profile at $T = 300$ K in 6 nm and 8.5 nm thick samples, respectively, at $t = 0.02$ ns neutron Fourier time. Strong signal-to-background ratio is detected in the NSE signal. **d** The echo becomes weaker at higher Fourier time ($t = 0.26$ ns), indicating the dominance of charge defect $q_m$ relaxation at a lower time scale. **e, f** Spin echo profile at $T = 4$ K in $h = 6$ nm and 8.5 nm thick samples, respectively, at $t = 0.02$ ns neutron Fourier time. Self-propelled charge dynamics prevails at low temperatures. In all plots, the error bar represents one standard deviation.

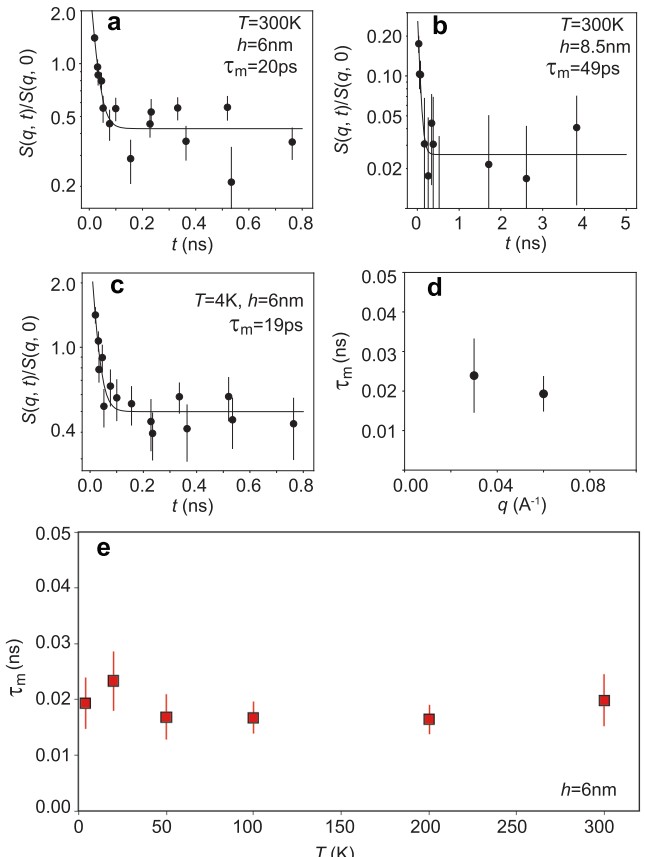

**Fig. 3 | Temperature dependence of magnetic charge defect's relaxation time. a, b** $S(q,t)/S(q,0)$ versus neutron Fourier time at $T = 300$ K for the $h = 6$ nm sample (**a**) and the $h = 8.5$ nm sample (**b**). The magnetic charge relaxation time $\tau_m$ is obtained from an exponential fit of the data. **c** Same as **a**, but for $T = 4$ K. **d** The plot of $\tau_m$ vs. $q$. **e** $\tau_m$ vs. $T$ in the 6 nm sample. Charge defect $q_m$ relaxes at $\tau_m$ -20 ps timescale at $T = 4$ K without any external stimuli. The plot also shows the temperature independence of $\tau_m$. In all plots, the error bar is calculated from the square root of the variance of the least-square-fitted parameter.

temperature[27]. The normalized intensity reduces to the background level above the spin echo Fourier time of -0.5 ns, conforming to the most general relaxation process signature in NSE measurements[27,28]. The quantitative determination of the magnetic charge relaxation time involves exponential fitting of the NSE scattering intensity as $S(q,t)/S(q,0) = Ce^{-t/\tau_m}$, where $C$ is a constant[29]. The exponential function provides a good description of the relaxation mechanism of charge $q_m$.

Four important conclusions can be drawn from the estimated value of the relaxation time: first, the obtained value of $\tau_m = 20$ ps at $T = 300$ K suggests a highly active kinetic state in our permalloy honeycomb lattice, typically not observed in nanostructured materials where the domain wall motion is the primary mechanism behind the dynamic property[30]. Moreover, a magnetic field or current application is necessary to trigger domain wall dynamics, which is not the case here: the dynamics state of charge $q_m$ is entirely self-propelled. Second, the relaxation rate of $q_m$ in the honeycomb lattice is somewhat thickness independent. In Fig. 3b, we show $S(q,t)/S(q,0)$ as a function of neutron Fourier time in the 8.5 nm thick lattice. The estimated $\tau_m = 49(10)$ ps is comparable to that found in the 6 nm thick lattice (within 2$\sigma$). Importantly, we do not observe a drastic difference between the charge dynamics in the thinner and the thicker lattice, which is a strong indication of its quasi-particle characteristic.

Third, the charge defect $q_m$ relaxes at the same high rate at low $T$ despite the absence of thermal fluctuation. As shown in Fig. 3c, the

estimated relaxation time as a function of temperature suggests the persistence of charge dynamics at lower temperatures. Given the fact that the charge dynamics prevail in the absence of any external stimuli, the observed $T$-independence is quite significant. The conclusion is that the system lives in a persistent kinetic state due to the magnetic charge defect motion between parent lattice vertices, which is strong evidence for the quantum nature of magnetic charge quasi-particles.

Lastly, the $q_m$ relaxation rate in the 6 nm thick honeycomb lattice is comparable to that found in bulk spin ice material, -5 ps[18,28]. This suggests a correspondence between the dynamic behavior of magnetic charge defects and the monopolar dynamics in the atomistic spin ice. To our knowledge, this is the only observation of its kind, bridging the gap between the bulk spin ice material and the nanostructured spin ice system. Thus, the quasi-particle characteristic of the monopole can be suitably invoked in the artificial lattice. These conclusions are further supported by theoretical analysis, which we now briefly summarize.

**Dynamic Monte-Carlo and classical micromagnetic simulations**
In order to confirm the quantum nature of magnetic charge defect quasi-particles, we modeled the system both fully quantum mechanically and fully classically. Motivated by the simplistic behavior of magnetic charge relaxation, unchanged by temperature or system size, we suspect the system exhibits rapid renormalization group convergence to a minimal effective model of behavior, and as such, we

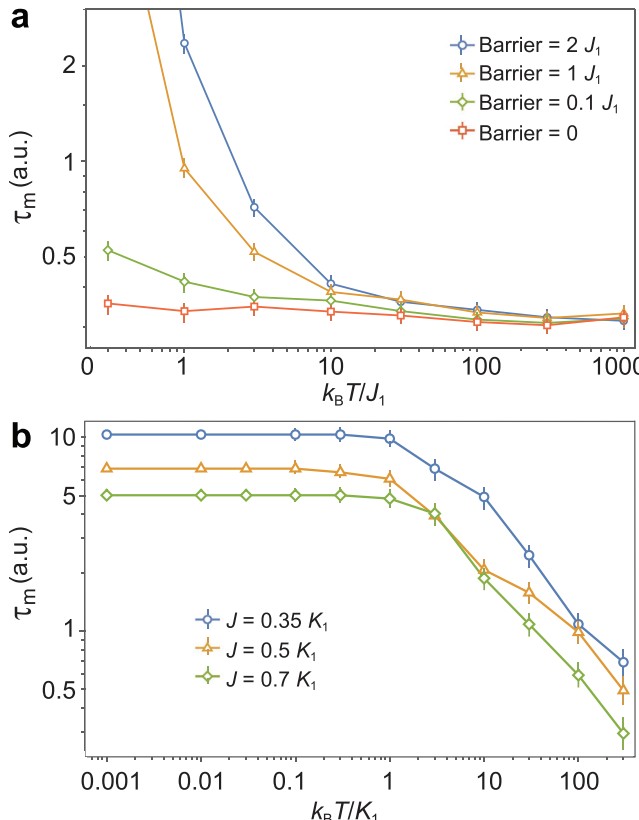

**Fig. 4 | Quantum mechanical nature of $q_m$ relaxation process. a, b** Temperature dependence of $\tau_m$ extracted from quantum (**a**) and classical (**b**) numerical simulations. In quantum simulations, the temperature is defined in the units of the nearest neighbor exchange constant $J_1$. In classical simulations, the temperature is defined in the unit of the effective anisotropy constant $K_1$, with $J$ denoting the nearest neighbor exchange constant. Only the dynamic Monte-Carlo simulations with negligible barrier height, compared to the exchange constant, result in temperature independence of $\tau_m$, as found in experimental data. It suggests effortless relaxation of $q_m$, similar to the quantum entity. The data points represent the mean from multiple runs of stochastic simulations, and the error bars represent the standard deviation of the mean.

assume the system should be modeled effectively by a Hamiltonian with minimal features. To this end, for the quantum modeling, we treat the magnetic monopoles at each vertex as a pseudospin 3/2 due to the quartet nature of these sites. These pseudospins are coupled minimally via nearest neighbor flip flop terms, i.e., $H_{nn} = \sum_{\langle ij \rangle} J_h S_i^+ S_j^-$ where $S_i^+$ is the spin 3/2 representation of an SU(2) raising operator for the $i$th site, and the sum is over nearest neighbors. We calculated the average relaxation time as a function of temperature for two cases, a 4-site model solved exactly, and a 200-site model with periodic boundary conditions time evolved via dynamic Monte Carlo[31]. Both cases reproduce the temperature independence of relaxation time for negligible barrier height compared to the large exchange constant, see Fig. 4a (see further details in Supplementary Notes C).

The main result of the dynamic Monte-Carlo simulations is that a magnetic charge defect $q_m$ travels through the lattice element effortlessly as a result of a negligible energy barrier. A similar observation was made in the case of magnetic monopole dynamics in atomic spin ice[5]. If the barrier height were comparable to the exchange constant $J$, then the relaxation time would increase significantly at low temperatures; however, this is not seen in the experimental data. Similarly, if the kinetic behavior of $q_m$ were classical in origin, then the simulation would predict a monotonic temperature dependence of the relaxation time above the barrier height.

For the classical modeling, we have treated each joint of the lattice as a classical spin obeying the Landau–Lifshitz–Gilbert equation with a stochastic field modeling temperature dependence[32]. The thermal effect is introduced using the method described by Leliaert et al.[32]. Since magnetic dynamics takes place on ~ps time scale, which is much larger than the correlation time in the thermal fluctuations (typically of the order of $10^{-13}$ s), we find the method to be a highly effective approximation. In the classical modeling case, the magnetic charge relaxation time was found to be temperature independent only up to a threshold temperature $T_c$, above which the relaxation time decreases monotonically with increasing temperature. This threshold occurs when the root-mean-square of the thermal field $B_{th}$ is comparable in magnitude to the effective field term associated with shape anisotropy $B_{an}$. With an estimated effective anisotropy (mostly shape anisotropy) of $|K_1| \sim 1 \times 10^4$ J/m$^3$, we estimate the threshold temperature to be on the order of $T_c \sim 20$ K, see Fig. 4b. This is inconsistent with the experimental observations, leading us to conclude that the charge defect $q_m$ dynamics has indeed quantum nature.

## Discussion

This is the first time that a comprehensive study of magnetic charge relaxation processes in a two-dimensional artificial permalloy honeycomb lattice reveals kinetic events with ~ps temporal quantization. Previous efforts in determining the charge dynamics in artificial spin ice systems focused on the usage of ferromagnetic resonance spectroscopy and optical measurements such as Brillouin light scattering and optical cavity techniques[33,34]. However, these techniques have intrinsic temporal thresholds of 1 ns or higher, which limits the scope of sub-ns investigation of charge dynamics. Additionally, the samples used for the dynamic study were created via electron-beam lithography, which results in a large element size. In most cases, samples created in this way are athermal[9], thus significantly affecting the spatially quantized relaxation process between the vertices of the parent lattice. Our honeycomb sample with small inter-elemental energy, utilized in this study, provided an archetypal platform to extract the charge relaxation properties in the absence of thermal fluctuation. The ultra-small element size imparts a thermally tunable characteristic to the lattice. It is also relevant to mention that the macroscopic size honeycomb specimen used in this study tends to develop structural domains that vary in size from 400 nm to a few μm. However, the NSE study of charge dynamics only involves the nearest and the next nearest neighboring vertices. Therefore, structural domains are not expected to affect the outcome. The NSE technique, with a dynamic range extending to sub-ns timescales, is the most suitable experimental probe to study such delicate dynamic properties.

Our combined experimental and theoretical investigations have not only revealed the sub-20 ps relaxation time associated with magnonic charge dynamics but also elucidated its quantum mechanical characteristics. The two observations—persistent dynamics in the absence of any external tuning agent and the near barrier-less transport inferred from dynamic Monte Carlo calculations—suggest the quantum mechanical nature of charged particle dynamics. The thickness independence of the relaxation time $\tau_m$ gives further support to the quasi-particle nature of the charge $q_m$. Despite the classical characteristic of magnetic charge, directly related to magnetic moment $M$ along the honeycomb element length ($l$) via $Q = M/l$, the observed quantum mechanical nature of the charge defect dynamics is unprecedented. Unlike magnetic monopole or magnon particles in bulk magnetic materials of atomic origin, $q_m$ is expected to possess finite macroscopic size. However, the kinetic behavior of $q_m$ manifests similarities with the known properties of magnetic monopoles, which is a strong indication of the particle-type character. Alternatively, the observed dynamic behavior may have its origin in the ring exchange mechanism, as claimed in quantum spin ice (QSI) material[35–38]. Unlike the ring created due to the successive flipping of spins on a diamond

lattice, forming a hexagonal loop in QSI, the intrinsic hexagonal unit cell in a honeycomb structure provides an ideal setup for such an event to occur. However, the current NSE spectrometer cannot access the low-$q$ regime needed to probe the ring exchange mechanism in our system. This will be an interesting subject for future investigations. Also, future studies using other experimental techniques, such as magnetic noise measurements and X-ray photon correlation spectroscopy, are desirable to further explore the observed phenomena.

The findings reported here can be utilized for the exploration of 'magnetricity' in artificial kagome ice, originally envisaged in the spin ice magnet[6]. Additionally, the quantum nature of magnonic charge can spur spintronic application via the indirect coupling with electric charge carriers[39]. In recent times, experimental efforts have been made to develop a spintronic venue in an artificial spin ice system[40]. A dynamic system of thermally tunable magnetic honeycomb lattices renders a promising research platform in this regard.

## Methods

### Sample fabrication
Artificial honeycomb lattices were created using hierarchical top-down nanofabrication involving diblock copolymer templates (see details in Supplementary Methods A). An atomic force micrograph of the honeycomb lattice sample is shown in Fig. 1d.

### Neutron spin echo measurements
We performed NSE measurements on permalloy honeycomb lattice samples using an ultrahigh-resolution (2 neV) neutron spectrometer SNS-NSE at beamline BL-15 of the Spallation Neutron Source, Oak Ridge National Laboratory. Time-of-flight experiments were performed using a neutron wavelength range of 3.5–6.5 Å and neutron Fourier times between 0.06 and 1 ns. NSE is a quasi-elastic technique where the relaxation of a magnetic specimen is decoded by measuring the relative polarization change of the scattered neutron via the change in the phase current at a given Fourier time (related to neutron precession). The experiment was carried out in a modified instrumental configuration of the NSE spectrometer where magnetism in the sample is used as a $\pi$ flipper to apply 180° neutron spin inversion instead of utilizing a flipper before the sample. Additional magnetic coils were installed to enable the xyz neutron polarization analysis; see the schematic in Supplementary Fig. S3. Such modifications ensure that the detected signal is magnetic in origin. NSE measurements were performed on two different parallel stacks of 125 and 117 samples of 20 × 20 mm surface size and 6 nm and 8.5 nm thicknesses, respectively, to obtain a good signal-to-background ratio. Considering the macroscopic separation distance between the honeycomb layers due to the 0.28 mm thick single crystal silicon substrate, it is not likely to have inter-layer coupling between the stacked honeycomb samples. The sample stacks were loaded in the custom-made sample cell, which was mounted in a close-cycle refrigerator with a 4 K base temperature. NSE data was collected for 8 hrs on average at each temperature and neutron Fourier time.

Minimum datasets that are necessary to interpret, verify and extend the research can be available from the corresponding author upon request.

## Data availability
The raw data that supports the findings of the study is stored in the central data repository at SNS-ORNL. The data can be made available upon reasonable request to the corresponding author.

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

## Acknowledgements

We thank Valeria Lauter and George Yumnam for helping us understand the magnetic charge distribution on honeycomb vertices. We also thank Antonio Faraone and Georg Ehlers for the helpful discussion on the use of the NSE technique in probing magnetic materials and designing the experiment. This work was supported by the U.S. Department of Energy, Office of Science, Basic Energy Sciences under Awards No. DE-SC0014461 (DKS) and DE-SC0019109 (CAU). This work utilized the facilities supported by the Office of Basic Energy Sciences, US Department of Energy.

## Author contributions

D.K.S and C.A.U. jointly led the research. D.K.S. envisaged the research idea and supervised every aspect of the experimental research. C.A.U supervised theoretical research. Samples were synthesized by J.G. and P.G. Neutron spin echo measurements were carried out by J.G., Y.C., L.S., P.Z. Analysis was carried out by J.G., L.S., P.Z., and D.K.S. Theoretical calculations were carried out by D.H. and C.A.U. The paper was written by D.K.S. and C.A.U. with input from all co-authors.

## Competing interests

The authors declare no competing interests.
