## [Peer Review File · Nature Communications]

Reviewers' Comments:

Reviewer #1:

Remarks to the Author:

This paper presents experimental results reporting a persistent dynamical state in an artificial spin ice system. The particularity of this system is that the metamagnetic length scale is less than 10nm, much smaller than existing experimental artificial spin ice systems. The observation of persistent dynamics is certainly interesting and the identification of an artificial spin ice system with measurable dynamics allowing for access to a real or effective thermodynamic states would certainly be a big breakthrough in the field. However, as it stands there are a number of issues that exclude it from publication in its present form.

1. The discussion of monopole dynamics and the analogy with microscopic spin ice materials is only partially correct. The authors definition of magnetic monopoles is incorrect. They state that "A magnetic moment can be thought of as a pair of '+' and '-' quasi-particles, called magnetic charges, coupled via magnetic Coulomb interactions⁴". These are the charges of a "dumbbell" and they are separated by a fixed length. The objects that interact via a Coulomb interaction are the magnetic monopoles of charge $2Q$ (as stated). In monopole dynamics, these objects and only these objects move through the system. Figure 1b is therefore very misleading - what we see here is a dipole (a dumbbell) and the operation is a flip in orientation of the dumbbell.

2. What the authors mean by quantum dynamics is also very confusing. Concerning spin ice materials HTO and DTO, these are known as "classical spin ices". That is, while there is a consensus that spins flip via a quantum tunnelling mechanism, the quantum mechanics stops here. The relaxation of the spin configurations is due to incoherent hopping of monopoles. There is an energy cost to create monopoles so that at low temperature, even though the monopole hopping rate can be considered as temperature independent, the relaxation time diverges and dynamics become very slow. The dynamics proposed here appears to be very different.

3. Within a spin ice picture, temperature independent relaxation dynamics can only come from coherent, monopole free dynamics such as ring exchanges. This would require coherent evolution on a length scale of 20 nm. In the monopole picture this would be quantum mechanical, otherwise it would need a classical low energy pathway outside the scope of this picture. If any of this this were the case then it would surely be interesting but it needs to be discussed.

4. Related to point 3, it is stated that creation of a $-3Q$ vertex, which is the creation of a monopole anti-monopole pair requires an energy of 76K. If the dynamics is due to monopoles and the system is able to exchange energy with the heat bath then the dynamics should be controlled by this energy scale yet is clearly is not. Are the authors sure that there is energy exchange between the heat bath and the meta magnetic degrees of freedom ? Could it be that the system of metamagnets is trapped in a high energy state with a fixed monopole concentration? Can they monitor the monopole concentration vs temperature through atomic force microscopy measurements or otherwise ? This would be a key measurement.

5. The authors need to be clearer about the modelling techniques used. I believe that the dynamical MC technique is a semi-classical technique so they are not doing full quantum Monte Carlo simulations as claimed on page 3. In addition, the effective Hamiltonian used only has off-diagonal terms so there is no potential energy that maintains the ice rules on a given vertex. As a consequence it is not surprising that this Hamiltonian gives a dynamic state even in a semi-classical regime. So, how can the authors justify justify this within the monopole picture ?

In conclusion, this is a very interesting experimental system but I do not find the interpretation in terms of quantum dynamics of spin ice systems very convincing. I believe that at the minimum the above questions should be addressed or refuted before the paper can be considered for publication.

Reviewer #2:

Remarks to the Author:

Domain wall kinetics in an artificial honeycomb lattice is an important current topic, with reference both to fundamental quantum systems and potential application in spintronics.

The sample preparation and the theoretical calculations presented in the manuscript are at a very high level.

The data quality looks poor, and my concern is whether these data are sufficient to support the conclusions. The experiment is at the limit of what is possible with the neutron spin-echo technique. The sample mass is tiny. In addition, the experiment is far more complicated than a standard spin-echo measurement. To the best of my knowledge, there are not many examples on polarization analysis on ferromagnetic samples without using additional polarizers. The data analysis and background treatment are crucial in this demanding experiment, and the supplement is not sufficient to reproduce the analysis procedure.

At present, I can't support publication. I ask the authors to give detailed answers to my questions on the experimental aspects.

1. What is the thickness of the substrate? Is there cross talk between the honeycomb layers in the stack?
2. How is the sample oriented with respect to the scattering plane?
3. There is probably a guide field at the sample. Will this magnetize the permalloy or give a bias in the domain orientation? Will this affect the relaxation dynamics?
4. Are the additional coils for the polarization analysis Helmholtz-style coils to rotate the guide field and are there also spin flippers? How are then the intensities z_{up} , $Z+$, $Z-$ etc. obtained?
5. Is there elastic signal contamination? The Permalloy volume is tiny, 0.1mm^3 (if I did the calculation right). Which signal/background ratio do you get before correction subtracting the data?

First, there should be no elastic SANS from the Permalloy honeycomb, this means peaks at the same Q 's (0.029 and 0.058 inv-Angström) corresponding to the dominant length scales L (leg) and $2L$ (diameter of cell).

Second, the same honeycomb structure is also present in the Si substrate. What is the depth of these structures in the Si?

Let us for the moment assume, that such elastic signals exist. Could they result in spurious magnetic signals in the polarization analysis, if the polarization efficiency is <1 , or if the calibration of the axes x,y,z has angular offsets?

6. The Q -assignment of pixels in Fig. S3 is only right, if the energy change is sufficiently small, otherwise one pixel could correspond to a broad range of Q . You mention an energy threshold of 20K ($\sim 1.7\text{meV}$) for the domain reorientation. Do you see inelastic scattering at energies beyond the spin echo resolution? Did you do TOF without spin-echo to confirm, that there is no inelastic scattering beyond the spin echo resolution range? If Q varies within one pixel, then the polarization analysis also might be affected, as the spin flip processes all refer to the orientation of Q . This also might induce spurious magnetic signals.
7. Are the spin-echo data in Fig 2b-f raw data obtained by the summation over detector x,y , time pixels, or is there more analysis involved? Why are, for example, the error bars in d,e much larger as in c , although the intensities are nearly equal?
8. How many time pixels were summed? What is then the mean λ and the effective λ band for the sum?

9. How is τ varied? By taking different time channels at constant field integrals, or by variation of the field integrals?

10. Why is the intensity for the 8.5nm sample in Fig. 2f much higher than in 2c?

11. In Fig. 2, there is a large phase shift between comparable configurations c and f (8.5nm, 20ps, 300K and 4K) but much less between b and c. For quasi-elastic scattering, there should be no phase shift. What is the reason here? A phase instability of the instrument? Please show more of these binned raw data. Are these phase shifts understood?

12. There are ferromagnetic domains randomly oriented along the legs of the structure. A neutron passing these domains will acquire significant precession phase or depolarization by passing these domains, where this effect depends on the trajectory of the neutron. Can you calculate or estimate this effect in your samples? It also might be a source for spurious magnetic signals in the polarization analysis.

13. Might the rapid decay of $S(q,t)$ at small τ (Fig 3) result from elastic contamination of your signal, rather than from quasi-elastic scattering from magnetic domain dynamics?

14. Why does $S(q,t)$ in Fig 3a stay constant at ~ 0.5 for $\tau > 0.1$ ns?

15. Why is $S(q,t)$ so much smaller in Fig. 2b.

16. The statement at the end of supplement page 5 (" Thus, the instrument resolution function is not divided out, ...") needs more detailed discussion. In addition, at small τ the simple relation in supplement eq(1) is not valid, as second order terms in the phase= $\omega \tau$ become relevant. Are you already in this regime in your experiment?

Reviewer #3:

Remarks to the Author:

I found this paper very difficult to referee and I feel my only option is to be completely transparent. At its core this manuscript is telling me something that I find very hard to believe on the basis of experimental data I don't fully understand. If the authors are correct it is a very significant result which will attract broad interest given the implications for data storage, extension of the quantum regime for quantum computing. Having said that I shall do my best.

The paper relates to a novel nanofabricated material with a honeycomb structure of ferromagnet. This has clear parallels with the artificial spin ice (ASI) family of materials that have been widely studied over the last 15 years, where a model of a single domain ferromagnet as a 'macrospin' or a dumbbell of magnetic charge, and magnetic reversals in a frustrated (e.g. kagome) lattice occurs by 1D chains behind a mobile magnetic charge excitation (monopole) have been hugely successful in explaining the experimental observations. However ASI typically has bar lengths of 100s of nm compared to the exchange length of ~ 5 nm, so the physics is classical dipolar and well described by the micromagnetic approximation, and there is in fact surprising little difference between the magnetic properties of connected and disconnects array of bars even though one has exchange coupling and the other does not. Most ASI structures are 'athermal' meaning the macrospin direction does not thermally fluctuate, although there are magnon precessions around the average direction. There has been a significant body of work where the nanomagnets are superparamagnetic meaning that above a blocking temperature there are thermal fluctuations but there is a clear Arrhenius behaviour with thermal activation. This manuscript pertains to a honeycomb nanostructure where the individual nanobars are $\sim 10\text{nm} \times 5\text{nm} \times 5\text{nm}$ (or $2 \times 1 \times 1$ exchange lengths). The central claim here is that the authors observe a temperature independent quantum fluctuation of the macrospins in neutron spin echo experiments. The very clear inspiration is the seminal work of Georg Ehlers at Oak Ridge with Steve Bramwell at UCL and collaborators, using Neutron spin echo to demonstrate a temperature independent plateau in the relaxation rate of the atomic spins of rare earth atoms in the pyrochlore spin ice crystals as proof of the "Coulomb phase" with quantum tunnelling of atomic spins enabling the magnetic monopole flow. A couple of these papers are cited (refs 27 and 28) though there are more. However there

are a number of ways in which the results of Ehlers, Bramwell etc.. are more convincing than this paper. They use ac susceptibility as an independent confirmation, they plot the data in a different way that is easier to understand, show data with the thermal decay of the relaxation rate in the temperature regions both above and below the plateau (e.g fig 1 of ref 28) so it is convincing they can measure the plateau, their error bars are vastly smaller and have a lot less scatter in the 'plateau' and finally that the quantum fluctuation of the moment on a single rare earth ion is not that surprising. On the other hand Seagate, Western Digital etc.. clearly believe they can make hard disks out of ferromagnetic domains at the 10nm*5nm*5nm scale, and if those domains were able to quantum tunnel out of their data state I would have thought they would have noticed by now.

Doing my best to understand the data: in $S(q,t)$ the q dependence should tell you what sort of object is changing, while the t dependence gives you the rate of change. If the q dependence is flat and the t -dependence is temperature independent then that would be tunnelling of single spins, which is what was measured by Ehlers and Bramwell. I think tunnelling of islands would manifest by a strong q dependence and a temperature-independent T -dependence. Perhaps showing $S(q,t)$ vs q as well as vs t would help. As things stand the data in Fig 3a,b and c look a lot like what you would get from the measurement of a simple exponential decay where your measurement hits a noise or background floor.

Additionally micromagnetic simulations are presented. I am not sure if the micromagnetic approximation is even valid when your feature size is smaller than the exchange length. Looking at Fig 1c I can see that the cell size used is basically 1 atom, so I don't understand why you wouldn't use an atomistic magnetic simulator like Vampire instead of a micromagnetic approximation with no atoms, especially when the temperature dependence is so vital and micromagnetics cannot handle temperature e.g. phonons etc well.

In summary, the claims are certainly of a suitable significance for Nature Comms, but I am not sure the case has been made sufficiently convincingly to justify such a surprising claim.

Reviewer #1:

This paper presents experimental results reporting a persistent dynamical state in an artificial spin ice system. The particularity of this system is that the metamagnetic length scale is less than 10 nm, much smaller than existing experimental artificial spin ice systems. The observation of persistent dynamics is certainly interesting and the identification of an artificial spin ice system with measurable dynamics allowing for access to a real or effective thermodynamic states would certainly be a big breakthrough in the field. However, as it stands there are a number of issues that exclude it from publication in its present form.

We thank the referee for reviewing our manuscript, and for recognizing that our observations of persistent dynamics in artificial spin ice would be a big breakthrough in the field. Below, we will address the referee's concerns.

1. The discussion of monopole dynamics and the analogy with microscopic spin ice materials is only partially correct. The authors definition of magnetic monopoles is incorrect. They state that "A magnetic moment can be thought of as a pair of '+' and '-' quasi-particles, called magnetic charges, coupled via magnetic Coulomb interactions⁴". These are the charges of a "dumbbell" and they are separated by a fixed length. The objects that interact via a Coulomb interaction are the magnetic monopoles of charge $2Q$ (as stated). In monopole dynamics, these objects and only these objects move through the system. Figure 1b is therefore very misleading - what we see here is a dipole (a dumbbell) and the operation is a flip in orientation of the dumbbell.

The referee is right about the monopole creation in spin ice. However, unlike the bulk spin ice where the tetrahedron has zero net charge in equilibrium, the concept of magnetic charge is different in kagome spin ice. As described in some of the seminal papers and reviews (see ref. 20, 21, 22, 8, 9), the vertices in a honeycomb lattice are always occupied by net charges of $\pm Q$ or $\pm 3Q$ due to the local moment arrangement of 2-in and 1-out (or vice-versa) or all-in (or all-out), respectively. **The dipolar interaction between magnetic charges is approximated by Coulomb interaction, as discussed in ref. 20, 21, 22.**

The referee is also correct that only charge defects of magnitude $2Q$ move through the system. This is the case in artificial spin ice as well. As the monopole charge $2Q$ moves through a honeycomb element, the moment direction changes (as the total charges on two neighboring honeycomb vertices change due to the emission or absorption of charge defect $2Q$, also demonstrated by micromagnetic simulations in Fig. 1c). That being said, we understand the referee's concern. To avoid potential confusion, we have modified the schematic Fig. 1b by removing the middle panel.

2. What the authors mean by quantum dynamics is also very confusing. Concerning spin ice materials HTO and DTO, these are known as "classical spin ices". That is, while there is a consensus that spins flip via a quantum tunnelling mechanism, the quantum mechanics stops here. The relaxation of the spin configurations is due to incoherent hopping of monopoles. There is an energy cost to create monopoles so that at low temperature, even though the monopole hopping rate can be considered as temperature independent, the relaxation time diverges and dynamics become very slow. The dynamics proposed here appears to be very different.

The referee is right. The dynamic process slows down at low temperature in bulk spin ice because there is not enough thermal energy to create monopoles. This is not the case in artificial kagome or honeycomb ice where magnetic charges are inherent due to the local magnetic configuration (**also discussed in detail in ref. 22**). Thus, they are always present and can exhibit dynamic processes if the system is thermally tunable. This is what we have in our honeycomb lattice. So, the relaxation time of magnetic charge does not slow down, rather exhibits persistent behavior at low temperature where thermal fluctuation is negligible. Quantum (dynamic) Monte Carlo (QMC) simulations suggest that this behavior can be best

described by a (near) barrier-less relaxation of magnetic charge defects. These two observations – persistent dynamics in the absence of any external tuning agent and the barrier-less transport in QMC calculations – suggest the quantum mechanical nature of charge particle dynamics. We have added this point in the discussion section of the manuscript.

3. Within a spin ice picture, temperature independent relaxation dynamics can only come from coherent, monopole free dynamics such as ring exchanges. This would require coherent evolution on a length scale of 20 nm. In the monopole picture this would be quantum mechanical, otherwise it would need a classical low energy pathway outside the scope of this picture. If any of this this were the case then it would surely be interesting but it needs to be discussed.

We thank the referee for bringing up the ring exchange mechanism. Indeed, the observed dynamic behavior may have its origin in the ring exchange mechanism, as claimed in quantum spin ice (QSI) material. Unlike the ring created due to the successive flipping of spins on a diamond lattice, forming a hexagonal loop in QSI, the intrinsic hexagonal unit cell in a honeycomb structure provides an ideal setup for such event to occur. However, the current NSE spectrometer cannot access the low-Q regime needed to probe the ring exchange mechanism in our system. This will be an interesting subject for future investigations.

We have added the above passage at the end of the paper (right before the Methods section) and cited some relevant papers (ref. 35-38).

4. Related to point 3, it is stated that creation of a $-3Q$ vertex, which is the creation of a monopole anti-monopole pair requires an energy of 76K. If the dynamics is due to monopoles and the system is able to exchange energy with the heat bath then the dynamics should be controlled by this energy scale yet clearly is not. Are the authors sure that there is energy exchange between the heat bath and the meta magnetic degrees of freedom ? Could it be that the system of metamagnets is trapped in a high energy state with a fixed monopole concentration? Can they monitor the monopole concentration vs temperature through atomic force microscopy measurements or otherwise? This would be a key measurement.

The referee has many good questions here. The sample was loaded in a closed helium environment in a close-cycle refrigerator. So, it is not in contact with the surrounding environment (or heat bath). Also, at $T = 4$ K, there is very little thermal energy to exchange between the environment inside the cryostat and the sample.

Magnetic force microscopy has a resolution of 50 nm or higher. So, we cannot use this to magnetically image the sample that has a length scale of ~ 10 nm. **Instead, previously we have used spin polarized neutron reflectometry to deduce the magnetic charge pattern on honeycomb vertices at different temperatures (see Ref. 24).** We had found two things: first, the system manifests massive degeneracy at all measurement temperatures. Second, the charge pattern on honeycomb vertices is comprised of both $\pm Q$ and $\pm 3Q$ charges. Importantly, the charge configuration remains mostly unperturbed as temperature decreases to $T = 4$ K, which means that most of the $3Q$ charges are still present at low temperature. Since $3Q$ charges are accompanied by higher energy, they are relatively unstable. We think that these charges spur the dynamic process by emitting (or absorbing in the case of $-3Q$) monopole charge of $2Q$ magnitude. But then some other low energy entities (Q charge) also undergo transient transformation to higher multiplicity charge. Thus, the process continues. These points are discussed in the introduction section of the manuscript.

It is difficult to say whether the system is trapped in a high energy state with fixed monopole concentrations. But if we think of $\pm 3Q$ charges as the source of monopoles (based on energetic consideration as the system would strive to attend low energy state), then it may be possible; the density of $\pm 3Q$ charges does not change much as a function of reducing temperature (see Ref. 24). To clarify this, we

have added the sentence “The presence of energetic $+3Q$ charges at low temperature suggests that the system maintains a high energy state throughout” in the introduction section of the revised manuscript.

5. The authors need to be clearer about the modelling techniques used. I believe that the dynamical MC technique is a semi-classical technique so they are not doing full quantum Monte Carlo simulations as claimed on page 3. In addition, the effective Hamiltonian used only has off-diagonal terms so there is no potential energy that maintains the ice rules on a given vertex. As a consequence, it is not surprising that this Hamiltonian gives a dynamic state even in a semi-classical regime. So, how can the authors justify this within the monopole picture?

We are not sure why the referee describes the effective Hamiltonian as “only off-diagonal”. There may be a misunderstanding. The effective Hamiltonian has diagonal terms proportional to the sum of Q^2 , and these Q^2 terms can have values of 1 or 9, so the Hamiltonian does have a nontrivial diagonal that effectively counts the number of monopole excitations. These diagonal terms are accounted for in the Monte Carlo: a hopping event that obeys the spin ice rules is given a probability proportional to $\text{Exp}(-dE/T)$ where dE is the change in the total Q^2 energy for that event. These details on the dynamics modeling are explained in the Monte Carlo reference [Ref. 31: Adler S. B., Smith J. W. & Reimer J. A. J. Chem. Phys. 98, 7613 (1993)].

The referee is correct that the dynamic Monte Carlo method is semi-classical. The local dynamics of excitations in this Monte Carlo method matches that observed in our 4-site exact quantum ensemble model, so the Monte Carlo simulation mainly serves the purpose of studying the larger scale dynamics and confirming that the behavior on an extended honeycomb lattice structure does not differ from that observed in the small-scale model. All instances of “quantum Monte Carlo” in the paper have been changed to “dynamic Monte Carlo” to be more in line with the terminology used elsewhere in the paper.

In conclusion, this is a very interesting experimental system but I do not find the interpretation in terms of quantum dynamics of spin ice systems very convincing. I believe that at the minimum the above questions should be addressed or refuted before the paper can be considered for publication.

We appreciate the referee’s questions that have improved the clarity of our paper. We believe that we have answered all of the referee’s concerns and hope that our paper can now move forward.

Reviewer # 2:

Domain wall kinetics in an artificial honeycomb lattice is an important current topic, with reference both to fundamental quantum systems and potential application in spintronics.

The sample preparation and the theoretical calculations presented in the manuscript are at a very high level. The data quality looks poor, and my concern is whether these data are sufficient to support the conclusions. The experiment is at the limit of what is possible with the neutron spin-echo technique. The sample mass is tiny. In addition, the experiment is far more complicated than a standard spin-echo measurement. To the best of my knowledge, there are not many examples on polarization analysis on ferromagnetic samples without using additional polarizers. The data analysis and background treatment are crucial in this demanding experiment, and the supplement is not sufficient to reproduce the analysis procedure.

At present, I can't support publication. I ask the authors to give detailed answers to my questions on the experimental aspects.

Thank you for reviewing our manuscript and acknowledging the uniqueness of our experiment. The neutron spin echo technique is a very powerful probe to investigate the dynamic properties of materials. At present it is mostly used to study soft materials, but we feel that it has great potential to be more broadly useful. We hope that our experiment will pave the way for its future usage in the exploration of nanostructured and thin film magnets. Based on the referee's comments/suggestions, we have significantly revised the supplementary information. Our responses to the referee's comments are discussed below.

1. What is the thickness of the substrate? Is there cross talk between the honeycomb layers in the stack?

The substrate is single crystal Si of about 0.28 mm in thickness. Considering the macroscopic separation distance between the honeycomb layers, it's not likely to have inter-layer coupling between them. We have added this information in the Methods section of the manuscript.

2. How is the sample oriented with respect to the scattering plane?

The sample stack is mounted in the transmission geometry such that the neutron beam direction is parallel to the sample normal direction. We have added this information on page 2 in the Supplementary Information.

3. There is probably a guide field at the sample. Will this magnetize the permalloy or give a bias in the domain orientation? Will this affect the relaxation dynamics?

Unlike in the soft matter NSE where the echo signal and the normalization factor can be measured in the same instrument configuration, paramagnetic NSE requires separate measurements for the echo signal and the total magnetic scattering for the normalization purpose. In the echo signal measurement, the scattered beam intensity is measured as one scan through the phase current that introduces the field integral asymmetry, and no field is applied around the sample. The total magnetic scattering is measured in the xyz polarization analysis where a small guide field is applied around the sample.

We assume that the referee is referring to the polarization analysis here. The small guide field here is around 1 mT (10 Oe), whose purpose is to define the quantization direction for the neutron spin and maintain its polarization. Considering the strong exchange coupling between magnetic moments in the sample, the small field of 10 Oe field is very unlikely to affect the magnetization or bias the domain in our honeycomb lattice during the xyz polarization analysis, just like in other neutron measurements that utilize polarized neutrons. It also does not affect the relaxation dynamics since the relaxation events are probed using the echo signal measurement, which does not employ the guide field around the sample.

Also, it is worth noting that the size of honeycomb element (~ 11 nm) is a bit smaller than the typical permalloy domain size (~ 12 nm). So, domain wall is not possible in our honeycomb lattice. We have added this information on page 2 in the Supplementary Information.

4. *Are the additional coils for the polarization analysis Helmholtz-style coils to rotate the guide field and are there also spin flippers? How are then the intensities z_{up} , Z^+ , Z^- etc. obtained?*

While the coils are Helmholtz-style coils, they are not in the exact Helmholtz configurations. There are no spin flippers. (Magnetism in the sample is used to flip neutron polarization). There are three sets of coils orthogonally mounted around the sample position for X, Y and Z orientations, please see the figure attached (also see figure S1 in the supplementary material). The currents in the coils are tuned based on the seminal paper by Ehlers et al., *Rev. Sci. Instrum.* 84, 093901 (2013). The polarization analysis is performed according to the method described in Chapter 11: *NSE spectroscopy and magnetism* by C. Pappas in the book of *Neutron Scattering from Magnetic Materials* 1st Edition, also mentioned by Ehlers et al. in *Rev. Sci. Instrum.* 84, 093901 (2013). (We have added this into the reference list in Supplementary Information). Here, Z_{up} and Z_{down} are measured as the spin-non-flip scattering (SNF) and the spin-flip scattering (SF) cross sections with the sample field along the positive Z direction. We have added this information on page 2 of the Supplementary Information together with the following Fig. S2 depicting the coils' set-up.

Fig: The set-up of the three sets orthogonally mounted X, Y and Z coils in the polarization analysis. The figure is added in the Supplementary Materials.

5. *Is there elastic signal contamination? The Permalloy volume is tiny, 0.1mm^3 (if I did the calculation right). Which signal/background ratio do you get before correction subtracting the data?*

We assume that by “*elastic signal contamination*”, the referee is referring to the instrument resolution function that is often identified as the elastic scattering below a frozen temperature in soft matter application of NSE. In this work, persistent dynamics is detected to the lowest measurement temperature. Thus, it is not suitable to use the lowest temperature data as the instrument resolution. Nevertheless, as we have discussed in the revised supplementary material, the instrument resolution does not change below 5 ns, see Fig. S9 (which is the entire Fourier time range of our experiment). Therefore, the resolution correction only introduces a net shift in the neutron intensity along the y-axis and does not affect the fitting of data to estimate the relaxation time, which is of most scientific importance. (This is a good question by the referee. We have added this brief discussion on pages 6-8 in the Supplementary Information along with the resolution plot Fig. S9.)

The signal/background ratio before the subtraction is estimated using magnetic scattering/total scattering measured during the polarization analysis. In the meantime, in paramagnetic NSE, only magnetic

scattering creates an echo signal without a π flipper at the sample (See Fig 2). That is what we have. Thus, we can calculate the echo amplitude/echo constant background as another indication of the signal/background ratio. We have added this information on page 6 of the Supplementary Information.

First, there should be no elastic SANS from the Permalloy honeycomb, this means peaks at the same Q 's (0.029 and 0.058 inv-Angström) corresponding to the dominant length scales L (leg) and $2L$ (diameter of cell). Second, the same honeycomb structure is also present in the Si substrate. What is the depth of these structures in the Si?

The depth of the honeycomb structure is on average around 5 nm. We have included below a depth profile along a cut line on an atomic force microscopic image of the honeycomb structure in the Si substrate. We have added this figure in the Supplementary Information.

Fig: The depth profile along a cut line on an AFM image of the honeycomb structure in the Si substrate shows that the depth of the honeycomb structure is on average around 5 nm. We have also added this information on page 1 of the Supplementary Information together with this image as Fig. S1.

Let us for the moment assume that such elastic signals exist. Could they result in spurious magnetic signals in the polarization analysis, if the polarization efficiency is <1 , or if the calibration of the axes x,y,z has angular offsets?

We are not sure what the referee means by “spurious magnetic signals”. The polarization analysis allows a straightforward distinction between magnetic and nonmagnetic contributions including the background, even though the efficiency of the polarization is never 100%. The angular offset in the polarization analysis was minimized by using a magnetic needle to make sure the polarization field direction aligns to the x , y , and z orientations. We have mentioned this on page 2 in the supplemental Information.

6. The Q -assignment of pixels in Fig. S3 is only right, if the energy change is sufficiently small, otherwise one pixel could correspond to a broad range of Q . You mention an energy threshold of 20K (~ 1.7 meV) for the domain reorientation. Do you see inelastic scattering at energies beyond the spin echo resolution? Did you do TOF without spin-echo to confirm, that there is no inelastic scattering beyond the spin echo resolution range? If Q varies within one pixel, then the polarization analysis also might be affected, as the spin flip processes all refer to the orientation of Q . This also might induce spurious magnetic signals.

The referee has made many good comments here. The referee is right that for a 2D area detector with ToF, each pixel corresponds to a broad range of Q arising from the wavelength band of neutrons from the source. The NSE spectrometer detector at SNS-ORNL has 42 ToF time channels. Each time channel labels the wavelength of the collected neutrons, which defines the resolution of neutron wavelength. Suffice to say that at a given X , Y pixel, within a single time channel, the collected neutrons are assumed to have the same wavelength hence the same Q . In Fig. S5, we have presented the Q -assignment of the pixels within a single time channel for illustration purposes. We have added this information on page 2 of the Supplementary Information.

We did not see inelastic scattering at energies beyond the spin echo resolution. As pointed out by the referee, the domain reorientation energy threshold (~ 1.7 meV) is out of the resolution of the NSE

spectrometer (2.5 neV – 0.33 meV), so we did not expect to see inelastic scattering signals related to domain reorientation in the NSE experiment. We have not performed other ToF inelastic measurements on this system as the small sample mass in nanostructured samples limits the measurement options. As the referee can see, even the NSE experiment was a challenge. **It took us more than two years to get decent data to draw a meaningful conclusion.** We thank Georg Ehlers at SNS-ORNL for his guidance in designing the NSE experiment.

For the polarization measurements, the referee is again correct that Q varies within one pixel. However, the orientation of Q does not change, only the magnitude of Q varies due to the neutron wavelength distribution. Only the relative orientation of the Q and the neutron spin polarization matters in the polarization analysis. From this point of view, the theory of polarization analysis holds and the results are valid.

7. Are the spin-echo data in Fig 2b-f raw data obtained by the summation over detector x,y,time pixels, or is there more analysis involved? Why are, for example, the error bars in d,e much larger as in c, although the intensities are nearly equal?

The spin echo data in Fig 2b-f are raw data obtained by summation over detector pixels X and Y in one single time channel. The error bars in Fig 2d,e seemed larger than that in Fig 2c because the y range in Fig 2d,e is smaller than the y range in Fig 2c. We have revised this figure where the range of y-axis in Figs. 2c and 2d-e are comparable. It removes the appearance aberration.

8. How many time pixels were summed? What is then the mean lambda and the effective lambda band for the sum?

We assume that the referee is referring to the echo profiles presented in Fig 2. The echo profiles are obtained from a single time channel corresponding to Fourier time 0.02 ns. For the 8.5 nm sample the mean wavelength is 3.7 Angstrom with the effective wavelength band of 0.07 Angstrom corresponding to a detector area of 3 x 12 in X, Y directions; for the 6 nm sample, with mean wavelength is 4.6 Angstrom with the effective wavelength band of 0.07 Angstrom corresponding to a detector area of 5 x 18 in X, Y directions due to a broader scattering peak. We have added this information on page 4 in Supplementary Information.

9. How is tau varied? By taking different time channels at constant field integrals, or by variation of the field integrals?

We have used both. During the measurements, the nominal Fourier time (0.06 ns, 0.1 ns, 0.3 ns, 0.7 ns, 1.0 ns) was varied by setting different values for the field integrals. Each value corresponds to an independent measurement. During the data analysis, by grouping different time channels, one can extract additional information. We have added this information on page 5 in the Supplementary Information.

10. Why is the intensity for the 8.5nm sample in Fig. 2f much higher than in 2c?

For reasons related to the experiment (acquisition time), the total number of neutron counts is different in the two echoes, these are raw data without normalization to the neutron flux. However, in the data analysis, data were normalized with respect to neutron flux. We have added this information on page 5 in SI.

11. In Fig. 2, there is a large phase shift between comparable configurations c and f (8.5nm, 20ps, 300K and 4K) but much less between b and c. For quasi-elastic scattering, there should be no phase shift. What is the reason here? A phase instability of the instrument? Please show more of these binned raw data. Are these phase shifts understood?

At SNS-NSE, the phase is extremely stable. There was a mistake in the plotting of Fig. f, which we did not catch. We thank the referee for pointing it out. We have corrected it. Fig. f has somewhat similar phase as in Fig. c. We have also added more data from 8.5 nm sample on page 7 (Fig. S8) in the Supplementary Information, as suggested by the referee.

12. *There are ferromagnetic domains randomly oriented along the legs of the structure. A neutron passing these domains will acquire significant precession phase or depolarization by passing these domains, where this effect depends on the trajectory of the neutron. Can you calculate or estimate this effect in your samples? It also might be a source for spurious magnetic signals in the polarization analysis.*

By “ferromagnetic domains randomly oriented along the legs of the structure”, we assume that the referee is referring to the magnetic moments of single domain permalloy that align along the length of the honeycomb connecting element due to shape anisotropy. These magnetic moments, when presented together at the macroscopic sample, behaves collectively as a paramagnetic system, which serves as the basis that makes our paramagnetic NSE measurement feasible. In this work, we have removed the π flipper, and used our sample itself as a π flipper. This is because the effect of the neutron scattered from a single magnetic moment is that the component of the neutron spin polarization (P) perpendicular (P_{\perp}) to the scattering vector Q depolarizes, while the parallel component (P_{\parallel}) undergoes a spin flip. The latter, on average, can be further interpreted as the superposition of two component of equal magnitude amounts to be half the magnitude of P . The first component (P_{180}) is equivalent to a 180-degree phase shift of half P around the normal direction of the plan constructed by Q and P , and the other component (P_{echo}) is equivalent to a 180-degree precession of half P with P_{\perp} as the flipping axis. This P_{echo} severs just like a π flipper in a standard NSE set-up. And by removing the π flipper, we are certain that any other scattering effect that cannot invoke a P_{echo} , including P_{180} and structural scattering will not contribute to an echo signal but only a flat background. See Figure 4 in Chapter 11: *NSE spectroscopy and magnetism* by C. Pappas in *Neutron Scattering from Magnetic Materials* 1st Edition. The reference was added in the manuscript.

13. *Might the rapid decay of $S(q,t)$ at small tau (Fig 3) result from elastic contamination of your signal, rather than from quasi-elastic scattering from magnetic domain dynamics?*

The rapid decay of $S(q,t)$ at small Fourier time reflects the intrinsic dynamics in the sample and is not due to the elastic contamination of the signal. We have included more information in the supplementary information explaining that the elastic scattering and instrument resolution do not affect the relaxation of the data (see Figure S9).

14. *Why does $S(q,t)$ in Fig 3a stay constant at ~ 0.5 for $\tau > 0.1$ ns?*

The $S(q,t)$ vanishes into incoherent scattering, and reaches the paramagnetic relaxation plateau beyond 0.1 ns Fourier time.

15. *Why is $S(q,t)$ so much smaller in Fig. 2b.*

We believe that the referee is referring to Fig. 3b. In Fig. 3a, b, the $S(q,t)$ are extracted from two distinct sets of samples with different overall masses that were measured at different times. The label on the y-axis is an arbitrary unit. The estimation of relaxation time is of most scientific importance here. We have added this information on page 8 of the Supplementary Information.

16. *The statement at the end of supplement page 5 (" Thus, the instrument resolution function is not divided out, ... ") needs more detailed discussion. In addition, at small tau the simple relation in supplement eq(1)*

*is not valid, as second order terms in the phase= ω * τ become relevant. Are you already in this regime in your experiment?*

We have expanded the discussion about the instrument resolution correction in the supplementary material to justify our decision of not applying the resolution correction to our results. The resolution is flat for the entire Fourier times used in this experiment (see Fig. S6 in Supp. Inf.). So, it doesn't affect the fitting as any resolution correction can only lift the intensity of entire dataset by a constant amount. For the second part of the question, we are still in the neV regime where the second order term is non-relevant. One uses the second order term when the fluctuation energy goes from neV to μ eV, which is not the case here. Equation (1) in the supplementary material is obtained from Chapter 1: *Fundamentals of Neutron Spin Echo Spectroscopy* by F. Mezei in *Neutron Spin Echo Spectroscopy, Basic, Trends, and Applications*. We have added this into the reference list in the Supplementary Information.

Reviewer #3:

I found this paper very difficult to referee and I feel my only option is to be completely transparent. At its core this manuscript is telling me something that I find very hard to believe on the basis of experimental data I don't fully understand. If the authors are correct it is a very significant result which will attract broad interest given the implications for data storage, extension of the quantum regime for quantum computing. Having said that I shall do my best.

The paper relates to a novel nanofabricated material with a honeycomb structure of ferromagnet. This has clear parallels with the artificial spin ice (ASI) family of materials that have been widely studied over the last 15 years, where a model of a single domain ferromagnet as a 'macrospin' or a dumbbell of magnetic charge, and magnetic reversals in a frustrated (e.g. kagome) lattice occurs by 1D chains behind a mobile magnetic charge excitation (monopole) have been hugely successful in explaining the experimental observations. However, ASI typically has bar lengths of 100s of nm compared to the exchange length of ~ 5 nm, so the physics is classical dipolar and well described by the micromagnetic approximation, and there is in fact surprising little difference between the magnetic properties of connected and disconnects array of bars even though one has exchange coupling and the other does not. Most ASI structures are 'athermal' meaning the macrospin direction does not thermally fluctuate, although there are magnon precessions around the average direction. There has been a significant body of work where the nanomagnets are superparamagnetic meaning that above a blocking temperature there are thermal fluctuations but there is a clear Arrhenius behaviour with thermal activation. This manuscript pertains to a honeycomb nanostructure where the individual nanobars are $\sim 10\text{nm} \times 5\text{nm} \times 5\text{nm}$ (or $2 \times 1 \times 1$ exchange lengths). The central claim here is that the authors observe a temperature independent quantum fluctuation of the macrospins in neutron spin echo experiments. The very clear inspiration is the seminal work of Georg Ehlers at Oak Ridge with Steve Bramwell at UCL and collaborators, using Neutron spin echo to demonstrate a temperature independent plateau in the relaxation rate of the atomic spins of rare earth atoms in the pyrochlore spin ice crystals as proof of the "Coulomb phase" with quantum tunnelling of atomic spins enabling the magnetic monopole flow. A couple of these papers are cited (refs 27 and 28) though there are more. However there are a number of ways in which the results of Ehlers, Bramwell etc.. are more convincing than this paper. They use ac susceptibility as an independent confirmation, they plot the data in a different way that is easier to understand, show data with the thermal decay of the relaxation rate in the temperature regions both above and below the plateau (e.g fig 1 of ref 28) so it is convincing they can measure the plateau, their error bars are vastly smaller and have a lot less scatter in the 'plateau' and finally that the quantum fluctuation of the moment on a single rare earth ion is not that surprising.

On the other hand Seagate, Western Digital etc.. clearly believe they can make hard disks out of ferromagnetic domains at the $10\text{nm} \times 5\text{nm} \times 5\text{nm}$ scale, and if those domains were able to quantum tunnel out of their data state I would have thought they would have noticed by now. Doing my best to understand the data: in $S(q,t)$ the q dependence should tell you what sort of object is changing, while the t dependence gives you the rate of change. If the q dependence is flat and the t -dependence is temperature independent then that would be tunnelling of single spins, which is what was measured by Ehlers and Bramwell. I think tunnelling of islands would manifest by a strong q dependence and a temperature-independent T -dependence. Perhaps showing $S(q,t)$ vs q as well as vs t would help. As things stand the data in Fig 3a,b and c look a lot like what you would get from the measurement of a simple exponential decay where your measurement hits a noise or background floor.

We appreciate that the referee has put our work in the context of prior research in the field of ASI. First of all, we would like to allay the referee's concern regarding the measurement limit. The NSE measurements at SNS were performed in typical Fourier times of 0.1 ns, 0.5 ns and 1 ns to estimate the fast relaxation rate of magnetic charges. In principle, the instrument at SNS can access even smaller Fourier time in the tune of ~ 5 ps. This has now been included on pages 2 and 5 in the Supplementary Information.

The referee has mentioned the name of Georg Ehlers. Indeed, we followed his suggestion in designing the experiment at SNS-ORNL. Based on his suggestion, we removed the spin flipper before the sample to ensure that the observed signal is coming from the intrinsic magnetism in the sample. So, NSE data truly reflects the dynamic property of honeycomb lattice system.

More recently, the main finding of persistent dynamic behavior in artificial honeycomb lattice is also independently confirmed by the preliminary coherent soft X-ray spectroscopy measurement (xpcs) at low temperature, albeit at longer time scale.

The figures above show Coherent soft X-ray spectroscopy pattern, obtained on permalloy honeycomb lattice at $T = 5$ K. The scattering patterns in Fig. a and b were collected at different times on the same spot size. One can see that the strength of dynamic correlation, represented by the color (yellow maximum), changes due to the intrinsic dynamic property. This is a preliminary result. Further experimental investigations are in progress.

Second, NSE is a quasi-elastic measurement technique, which requires large sample mass. The spin ice sample's mass in George and Steve's experiment was several grams. We have nanostructured material. No matter how many samples we stack, the overall mass will always be much smaller. So, it is not surprising that the data from our honeycomb lattice has a bigger error bar, but still exhibits good signal-to-background ratio. However, the error bar is relatively smaller in low Fourier time regime where it is more important for the estimation of relaxation time.

As an aside: There are not many examples of NSE investigation of magnetic dynamics in nanostructured materials. This is perhaps one of the very few successful experiments ever conducted on nanoscopic magnets in the past many years. We hope that our research paves the way for further use of this powerful probe to study nanomagnetic materials and thin films.

Third, the relaxation mechanism in artificial honeycomb ice is most likely different from the atomistic spin ice compound. Unlike the bulk spin ice material, where hopping and tunnelling of magnetic monopoles is the primary mechanism, the magnetic charge defect relaxes along the length of element in artificial spin ice. So, it is possible that they manifest different temperature and q -dependences. Only the relaxation time is comparable in the two systems (compared to the value at moderate temperature in spin ice). As the first referee has pointed out, the temperature independence of relaxation time in the artificial honeycomb lattice can also be related to the coherent ring exchanges in quantum spin ice. Although, we are not sure about the ring exchange mechanism in our system, but it shows that the phenomena we report here may have more exotic origin. In any case, our report is the first experimental investigation of magnetic charge relaxation in any artificial spin ice system. These points are highlighted in the discussion section of our paper.

Finally, we only observe NSE signal at two q values, corresponding to the length scale between the nearest neighbors and the next nearest neighbors. Due to the geometrical constraints (as the referee may already know), we could not measure at further lower q values. Comparing τ_m at the available q values, we infer that the relaxation pattern exhibits weak q dependence. We hope that the future measurements on new NSE spectrometers that are currently being built at ISS or in the process of being built at the second target

station (STS) at SNS can shed further light on this behavior. But the more important thing (also the main theme of the manuscript) is the observation of persistent dynamic behavior at low temperature without the application of any external tuning agent, which is a totally new finding. We have mentioned this several times in the paper, including in the discussion section.

Following the referee's suggestion, we have added a plot of q vs. τ_m in Fig. 3 in the revised manuscript.

Additionally, micromagnetic simulations are presented. I am not sure if the micromagnetic approximation is even valid when your feature size is smaller than the exchange length. Looking at Fig 1c I can see that the cell size used is basically 1 atom, so I don't understand why you wouldn't use an atomistic magnetic simulator like Vampire instead of a micromagnetic approximation with no atoms, especially when the temperature dependence is so vital and micromagnetics cannot handle temperature e.g. phonons etc well.

In response to the referee's question about the validity of micromagnetic approximation for feature sizes smaller than the exchange length, we contend that this is precisely the regime in which our approximation of the magnetic element as a single dipole is particularly valid. If the exchange length is larger than the feature size, then a large majority of the spins are ensured to be closely aligned within that feature; meaning that the spins together act effectively as a single spin. As such, we expect that a more detailed model would not change the highly intuitive result found with our model (specifically that the classical modeling predicts a temperature dependent change in the monopole hopping rate when the thermal energy becomes larger than the anisotropy energy).

In regard to modeling temperature, the method we used has been thoroughly validated in the cited work [Ref. 31: Leliaert J. et al. AIP Adv. 7, 125010 (2017)] and was found to be a highly effective approximation provided that the magnetic dynamics being modeled takes place on time scales much larger than the correlation times in the thermal fluctuations, which are on the order of 10^{-13} s. The dynamical features we are modeling take place on the order of 10^{-10} s. So, we believe the requisite conditions for this approximation are sufficiently satisfied. We have added this clarification on page 3 of the revised manuscript.

In summary, the claims are certainly of a suitable significance for Nature Comms, but I am not sure the case has been made sufficiently convincingly to justify such a surprising claim.

We are grateful to the referee for recognizing the significance of our claims, and hope that the referee is convinced by our explanations.

Reviewers' Comments:

Reviewer #1:

Remarks to the Author:

The

Reviewer #2:

Remarks to the Author:

The authors carefully answered my questions. The corresponding sections in the supplement were significantly extended and improved.

The revised manuscript and supplement give the impression of a well designed experiment. The data analysis and scientific interpretation are sound.

I now support publication in Nature Communications.

Reviewer #3:

Remarks to the Author:

I thank the authors for the time taken to respond to my questions. I believe these claims are too significant to underpin with a single technique that is little used for nanostructures, and the response has not convinced me of the certainty of the interpretation. I believe that the signatures of the claimed effect should show up in e.g. magnetic noise measurements, ac susceptibility. As such I cannot recommend for publication in Nature Comms at this time

Reviewer #3 (Remarks to the Author):

I thank the authors for the time taken to respond to my questions. I believe these claims are too significant to underpin with a single technique that is little used for nanostructures, and the response has not convinced me of the certainty of the interpretation. I believe that the signatures of the claimed effect should show up in e.g. magnetic noise measurements, ac susceptibility. As such I cannot recommend for publication in Nature Comms at this time.

Neutron spin echo (NSE) is one of the most powerful experimental techniques to study dynamic properties in a broad variety of materials, including nanomagnetic systems (see the paper by S. K. Sinha, Future science at next generation neutron sources, *Physica B* **356**, 269 (2005)). Additionally, we had provided preliminary results from X-ray Photon Correlational Spectroscopy (XPCS) measurements in our prior response that clearly showed the variation in dynamic pattern as a function of time. Thus, the phenomena can also be detected using other measurement techniques that are regularly employed to study artificial spin ice systems.

We agree that, in principle, magnetic noise measurements could be another possible approach to further understand the observed effect. However, this is beyond the scope of this work. Also, we don't know any research group who performs that type of measurement. We have added this possibility, along with XPCS technique, in the discussion section on page 5 (highlighted in red color) of the second revision of the manuscript for future study.

AC susceptibility measurements are not suitable to study the observed effect for two reasons: first, the ac frequency in commercially available Quantum Design PPMS is limited to 10^4 Hz, corresponding to the time scale of 0.1 ms. Given the fact that magnetic charges relax at ~ picosecond, this technique is not suitable to probe such fast dynamics. Second, the ac susceptibility measurement is a bulk technique. It requires a small sample (to be properly centered between the ac coils) with large enough mass (of the order of few g) to get good signal. Our sample is nanostructured system with very thin permalloy layer. So, it has very small mass and is therefore not suitable for ac susceptibility measurements.